# Assembly of Hollow Yttrium Oxide Spheres from Nano-Sized Yttrium Oxide for Advanced Passive Radiative Cooling Materials

**DOI:** 10.3390/polym16121636

**Published:** 2024-06-09

**Authors:** Jeehoon Yu, Daeyul Kwon, Heegyeom Jeon, Youngjae Yoo

**Affiliations:** Department of Advanced Materials Engineering, Chung-Ang University, Anseong 17546, Republic of Korea; yujeehoon@cau.ac.kr (J.Y.); koh1077@cau.ac.kr (D.K.);

**Keywords:** passive radiative cooling, polydimethylsiloxane, yttrium oxide, particle structure control, hollow particle, assembled nanoparticle

## Abstract

This study presents significant advancements in passive radiative cooling (PRC), achieved using assembled hollow yttrium oxide spherical particles (AHYOSPs). We developed PRC films with enhanced optical properties by synthesizing micro-sized hollow Y_2_O_3_ particles and integrating them into a polydimethylsiloxane (PDMS) matrix. The findings revealed that AHYOSPs achieved a remarkable solar reflectance of 73.72% and an emissivity of 91.75%, significantly outperforming nano-sized yttrium oxide (NYO) and baseline PDMS. Field tests demonstrated that the AHYOSPs maintained their lowest temperature during daylight, confirming their superior cooling efficiency. Additionally, theoretical calculations using MATLAB indicated that the cooling capacity of AHYOSPs reached 103.77 W/m^2^, representing a substantial improvement over NYO and robustly validating the proposed nanoparticle assembly strategy. These results highlight the potential of structurally controlled particles to revolutionize PRC technologies, thereby offering a path toward more energy-efficient and environmentally friendly cooling solutions.

## 1. Introduction

Under extreme indoor temperature conditions, such as during hot summers or cold winters, the energy derived from fossil fuels is often used to regulate and stabilize these environments, keeping them within a comfortable range [1]. Most of this energy, which is predominantly generated from fossil fuels, is used as electricity [2]. The thermal output from this process and the greenhouse gases emitted during combustion contribute to accelerating global warming [3]. In response, international treaties have been signed to curb these emissions, with researchers exploring methods to minimize fossil fuel energy use. A significant outcome of these efforts is the development of passive radiative cooling (PRC), a technique that cools without consuming additional energy [4]. PRC is versatile enough to reduce temperatures in apparel and residential spaces, such as buildings and vehicles, offering a cooling effect without reliance on external energy sources. The mechanism of PRC involves the transfer of thermal energy through electromagnetic radiation, directing it away from the Earth and into space, bypassing atmospheric absorption. For this process to be efficient, the heat radiated from external sources must be less than the expelled heat [5], which requires emissions to occur within specific wavelengths that the atmosphere does not absorb and necessitates the blocking of solar energy, a primary source of heat [6]. Prior studies have incorporated both organic and inorganic particles into polymer matrices to create composites and have validated the efficacy of this cooling approach [5]. The structural aspects of the materials utilized in PRC have also been comprehensively studied.

The field of PRC has grown considerably since the mid-2010s [7] with innovations including the introduction of emitters such as layered metal stacks [8], layers with hierarchical porosity [9], 2D nanopatterned layers [10], and nanofibers created via electrospinning [11]. In addition, modifying material types and forms has led to enhancements in the efficiency of PRC [12,13,14,15,16]. Further developments include crafting covers that inhibit thermal exchange between the emitter and environment [15,16]. When crafting PRC films, incorporating a significant volume of filler (over a 50 vol.%) into the emitter enhances the film’s cooling performance but complicates the production process due to the increased viscosity, making the blending of fillers and the creation of uniform films challenging [17]. Conversely, emitters with low filler content show reduced efficiency, necessitating the use of Appendix A. Previous studies have utilized glass bubbles (GBus) and barium sulfate (BaSO_4_), known for their theoretically high solar reflectance and radiative capabilities within the atmospheric window, as fillers for emitters [17,18].

Yttrium oxide (Y_2_O_3_) has been employed to achieve superior PRC efficiency. The refractive index of this material differs from that of its matrix, closely matching the emissivity of silica in the atmospheric window, and it is chemically stable. Utilizing hollow particles to construct a material structure increases the number and surface area of its reflective interfaces, enabling effective solar reflection [19,20]. In previous research, polydimethylsiloxane (PDMS) was chosen as the matrix for PRC films due to the electromagnetic radiation caused by bonding vibrations within its atmospheric window. PDMS, an inorganic polymer, features a Si-O-Si bond—pivotal in radiative cooling studies on emitting electromagnetic waves in the atmospheric window [21]. Among the fillers that excel in terms of their radiative cooling effects, silica, which possesses a Si-O-Si bond akin to PDMS, offers high-atmospheric-window emissivity and solar reflectance [22]. However, blending silica with PDMS could result in similar refractive indices because of their similar elemental contents, potentially lowering their solar reflectance. Therefore, Y_2_O_3_ was selected as an alternative due to its favorable atmospheric window emissivity and high solar reflectance [23].

In this study, Y_2_O_3_ was selected as the inorganic filler material due to its remarkable chemical stability and resistance to high temperatures. This selection is as we aim to sustain high radiative cooling efficiency over an extended period using an external coating. We synthesized hollow microparticles by assembling Y_2_O_3_ nanoparticles according to previous studies. We observed enhanced optical properties when transitioning from nanoparticles to microparticles and after adjusting the structure. This alignment with finite-difference time-domain (FDTD) simulation trends helped in predicting improvements in PRC performance. In addition, its cooling capabilities and effectiveness were confirmed via external experiments and compared with the intermediate particles created during the production process. The optical characteristics of the materials developed during the manufacturing stage were forecasted using FDTD simulations and compared with the properties of the ultimately manufactured particles, with their actual cooling performance and effectiveness substantiated through additional external experiments.

## 2. Experiments

### 2.1. Materials

All the particles used in this study were synthesized in the laboratory. PDMS (Sylgard 184, DOW Corning, Midland, MI, USA) was used as the film matrix and to evaluate the properties of the particles. Formic acid and NaOH (Samchun, Ansan-si, Republic of Korea) were used as reagents for particle fabrication. Melamine, formaldehyde, Y(NO_3_)_3_·6H_2_O, and urea were procured from Sigma-Aldrich (St. Louis, MO, USA). Deionized water (DI) served as the solvent for the aqueous reactions.

### 2.2. Fabrication of PRC Films

A core–shell structure coated with yttrium was initially created to synthesize the assembled hollow yttrium oxide spherical particles (AHYOSPs). The internal sacrificial template was removed, enabling the nanoparticles to assemble and form hollow structures. Melamine formaldehyde (MF) cores, used as the basis for the core–shell architecture, were developed as sacrificial templates. Initially, formaldehyde (0.36 mol), NaOH (10 wt%), and melamine (0.12 mol) were combined in deionized water and stirred at 50 °C for 40 min to create an MF prepolymer solution. The color of the solution changed from white to transparent. Subsequently, formic acid was introduced into the condensed water to produce the MF cores. The sizes of the MF cores can be adjusted by changing the experimental parameters. The resulting MF was purified using DI water and ethanol to eliminate small MF particles. The solvent was removed by vacuum-drying overnight at 60 °C; the MF cores were sealed and stored at ambient temperature. The core–shell structure was constructed using the produced MF; Y(NO_3_)_3_·6H_2_O, yttrium raw material, with MF as the core, and urea were mixed in DI water, and ultrasound was employed to disperse the MF. This well-dispersed solution reacted in a sealed environment at 85 °C for 3 h to produce the core–shell structure MF@Y(OH)CO_3_, with the unreacted MF and nano-sized MF@Y(OH)CO_3_ particles filtered out. The core–shell structure was stored under the same conditions as the MF core. For AHYOSP production, the prepared MF@Y(OH)CO_3_ core–shell structure was heated in a muffle furnace at a rate of 2 °C/min to a peak of 800 °C to eliminate the sacrificial template. The temperature was maintained for 2 h with oxygen added to the furnace, which caused the sacrificial template MF to burn and oxidize the shell structure into Y_2_O_3_. Nano Y_2_O_3_ (NYO) synthesis was performed without the MF core under similar conditions. To fabricate the PRC films, the fillers serving as reference samples during production were uniformly blended and stirred with PDMS at a concentration of 30 vol.% for each particle type (NYO and AHYOSPs). This mixture was then spread on a 2 mm thick glass Petri dish and cured under vacuum at 60 °C to form PRC films.

### 2.3. Characterization

A dynamic light scattering (DLS, LitesizerTM 500, Anton Paar, Graz, Austria) analysis was performed to assess changes in particle size distribution throughout the manufacturing procedure. Subsequently, scanning electron microscopy (SEM, Mira 3LMU FEG, Tescan) was employed to ascertain the particles’ morphology and composition. Furthermore, the optical properties crucial for radiative cooling were confirmed through UV–visible near-infrared (UV–Vis NIR, Cary 5000, Agilent Technologies, Santa Clara, CA, USA) and Fourier-Transform Infrared (FT-IR, Alpha-P, Bruker, Billerica, MA, USA) analyses. In the UV–Vis NIR analysis, the measurement range spanned from 250 nm to 2500 nm and encompassed the solar radiation spectrum. For the FT-IR analysis, the electromagnetic wave characteristics of the PRC film were examined, which necessitated the propagation of the material through the atmosphere. Notably, electromagnetic waves with wavelengths coinciding with the absorption bands of nitrogen and oxygen—the principal components of the atmosphere—cannot escape to outer space and are instead trapped within the Earth’s atmosphere.

### 2.4. Optical Properties Simulation

FDTD simulations, which serve as computational tools (Lumerical FDTD simulation program from Ansys—Canonsburg, PA, USA) for forecasting the optical attributes of these particles, were conducted to assess the optical characteristics of the synthesized particles. Leveraging these computational insights helps us determine the particle size pertinent to the solar wavelength spectrum—a pivotal parameter in effective radiative cooling. Additionally, these simulations facilitated the deriving of key optical metrics such as scattering efficiency, reflectance, and transmittance for the PRC film, serving as a valuable reference for producing tangible PRC particles. The FDTD simulations yielded electric field distributions, thus providing a visual representation of the wave energy propagated through the reflection and scattering phenomena.

### 2.5. Field Test

An external experiment was conducted to assess the cooling power of the PRC film in a real-world outdoor setting. The measurement site was Ansung, Chung-ang University (latitude 37°00′27.1″ N, longitude 127°13′48.1″ E), with the experiment conducted on 21 September. Figure 1 illustrates the configuration of the field test setup. The sides of the foam box were shielded with Mylar to minimize all other light sources apart from sunlight. The PRC film sample was positioned on top of the box to ensure it efficiently reflected and emitted sunlight. A thermocouple was then strategically placed beneath the sample to record the temperature fluctuations. Furthermore, for a comparative analysis of cooling performances under sunlight, a solar measuring instrument was deployed to measure the solar radiation levels. An anemometer and hygrometer were installed to provide additional context regarding the environmental conditions during the experiment. Moreover, the samples were shielded with high-transmittance PE films, specifically in the solar and atmospheric window regions, to regulate the external variables that could significantly influence the film’s cooling performance. This precaution aimed to mitigate the effects of heat conduction and convection during the experiments.

### 2.6. Cooling Power Calculation

The measurement of cooling power indicates the level of cooling achieved by the PRC film, which depends on the equilibrium between the incoming and outgoing thermal energy. In PRC calculations, energy transfer relies on the optical characteristics of the material. A black body is an ideal material that emits 100% across all wavelengths, with its radiance *I*_BB_ described by Planck’s equation, as follows:(1)$$I_{BB}=({2hc}^{2}/\lambda^{5})/\left[e^{hc/\left(\lambda \kappa_{B}T\right)}-1\right]$$

Here, h represents Planck’s constant, c the speed of light, λ the wavelength emitted by the black body, $\kappa_{B}$ the Boltzmann constant, and *T* the temperature of the sample. Using this equation, the radiative properties of the sample can be determined as
(2)$$P_{rad}\left(T\right)=A\int d\Omega cos\theta \int_{0}^{\infty }d\lambda I_{BB}\left(T,\lambda \right)\varepsilon \left(\lambda ,\theta \right)$$

In this formula, A is the sample’s area and $\varepsilon \left(\lambda ,\theta \right)$ is the emissivity of the PRC film, defined as $\varepsilon_{atm}\left(\lambda ,\theta \right)={1-t(\lambda )}^{1/cos\theta }$. t(λ) is the transmittance in the atmospheric window, calculated using ATRAN modeling software (Lord, S.D. 1992, NASA Technical Memor. 103957). Notably, non-radiative cooling losses such as conduction and convection are separate energy transfer mechanisms. The energy input via these pathways is expressed as
(3)$$P_{non,rad}=Ah_{c}\left(T_{amb}-T_{sam}\right)$$

Here, $h_{c}$ denotes the thermal coefficient, which changes according to the ambient temperature, humidity, and wind speed. The instantaneous thermal coefficient can be determined by substituting the measured temperature of the sample into the cooling power equation. The radiative energy influx into the sample included $P_{atm}(T_{amb})$ from the environment and $P_{Sun}$ from the Sun. Radiative energy from the surroundings is assumed to originate from the air, while solar energy reaches the sample through the atmosphere, covering the spectrum from 250 nm to 2.5 μm, encompassing the UV, visible, and near-infrared bands. The components are as follows:(4)$$P_{atm}(T_{amb})=A\int d\Omega cos\theta \int_{0}^{\infty }d\lambda I_{BB}(T_{amb},\lambda )\varepsilon (\lambda ,\theta )\varepsilon_{atm}(\lambda ,\theta )$$
(5)$$P_{Sun}=A\int_{0}^{\infty }d\lambda \varepsilon (\lambda ,\theta )I_{AM1.5}(\lambda )$$

In Equation (6), the passive radiant cooling power $P_{net}$, which consolidates the equations, is summarized as follows:(6)$$P_{net}(T)=P_{rad}(T)-P_{Sun}-P_{non,rad}-P_{atm}(T_{ambient})$$

To enhance the cooling power, reducing the external energy input and increasing the emitted energy level is imperative.

## 3. Results and Discussion

### 3.1. Assembling of Hollow Yttrium Oxide Spherical Particles

This research aims to determine whether the optical properties and PRC efficiency of yttrium oxide can be improved by assembling yttrium oxide nanoparticles into hollow microparticles, as shown in Figure 1 and Figure 2a. This assembly enhances their reflectivity and emissivity. Figure 2b,c provide insightful views of the well-synthesized AHYOSPs and the morphology of the nano-sized yttrium oxide (NYO) before assembly, confirming that well-developed nano-sized spherical particles form a structure with hollow particles. The AHYOSPs, with a size of 3.6 μm, and NYO, measuring 380 nm, were confirmed to be 1 μm and 11 μm, respectively, through DLS analysis (Figure 2f), indicating particle aggregation. SEM observations showed that the internal surface of the AHYOSPs appeared flat, likely shaped by the surface of the sacrificial template and the MF particles. The well-synthesized PRC particles were integrated at a 30 vol.% into the PDMS matrix, and, as observed in the cross-sectional images in Figure 2d,e and Appendix Aa,b, they were well-dispersed and embedded within the matrix.

### 3.2. Simulation of Optical Properties

We verified the morphology of the particles synthesized earlier and modeled the structure of the PRC particles based on their morphology. Therefore, PDMS was used as the matrix for our PRC film, with simulations conducted with precise spacing arrangements to ensure that the PRC filler particles were positioned at the central focal point. Figure 3a–c show the distribution of the electric field within the FDTD cross-section. A wavelength of 250 nm, chosen for the electric field distribution because this wavelength has the highest energy that comes from sunlight, facilitated the identification of unique characteristics, especially the backscattering occurring in the AHYOSPs compared to the unassembled nano-sized particles. In Figure 3a, the electric field distribution caused by scattering was not clearly observed; therefore, the resolution was changed to confirm scattering, indicating that the unassembled nanoparticles have a lower sunlight scattering efficiency than AHYOSPs. Moreover, at the longer wavelength of 1102.52 μm, the scattering of the controlled AHYOSPs is stronger than that of the nanoparticles, resulting in higher solar reflectance due to their higher total scattering efficiency in the sunlight-illuminated range compared to when not assembled. This simulation revealed the complex optical characteristics of PRC particles, shaping their performance in real-world applications.

### 3.3. Optical Properties of PRC Films

Figure 4a shows a reflectance graph, a key indicator for evaluating the cooling capacity of PRC films. In the spectral range of 250 nm to 2.5 μm, which corresponds to range of the solar radiation, high solar reflectance is directly linked to enhanced emissivity in the 2.5–20 μm range, where atmospheric window transmittance is dominant. Therefore, high solar reflectance amplifies the cooling power of these films. When calculating their average reflectance within the solar range, the AHYOSPs recorded the highest value of 73.72%, whereas NYO registered a value of 57.99%, thus confirming that the morphology of the assembled particles directly contributes to increasing their solar reflectance and plays a crucial role in shaping their optical properties. Figure 4b shows the films’ emissions within the atmospheric window. The AHYOSPs exhibited the highest emissivity, with an average of 91.75%. Considering these optical characteristics, we expect well-controlled assembled yttrium oxide to be used in PRC films to generate high atmospheric window radiation.

### 3.4. Field Test

This section empirically validates the cooling performance of the developed PRC films through field experiments. Figure 5a shows a schematic of the experimental setup used for data acquisition. Figure 5b shows the temperature variations within the samples, which correspond to the concurrent solar radiation data; a lower value indicates a more efficient cooling process. The close alignment of the solar radiation trends with the temperature fluctuations observed in the measured PRC films was noteworthy, proving that no external factors have influenced our observations and thereby confirming that the results are directly attributable to the effects of solar radiation. The shaded areas in the graph represent periods of reduced solar radiation at night. As shown in Figure 5c, the AHYOSPs consistently exhibited the lowest temperature during daylight. This thermal hierarchy directly corresponds to the solar reflectance characteristics of the materials and is primarily driven by their individual solar reflectance values. Importantly, the structural attributes of the particles also play a pivotal role in shaping the variations in solar reflection. The nocturnal trends differed from the daytime patterns, with a narrowing gap observed in cooling efficiency during the day, indicating that, as solar radiation decreases, emissivity contributes more dominantly to cooling than reflectance, with the temperatures of the ASYOSPs and NYO becoming almost identical as the night approaches. During the day, uncontrolled nano yttrium oxide showed approximately a 1 °C improvement in cooling efficiency and a 7 °C improvement in cooling efficiency compared to the PDMS baseline. Our PRC film performance enhancement demonstrates that there is potential to achieve improved cooling efficiency through particle control in PRC research.

### 3.5. Cooling Power Calculation

The theoretical net cooling power of the PRC film was estimated using MATLAB-based calculations to quantitatively investigate its cooling performance. To calculate its theoretical cooling capacity, we used Equations (1)–(6) and applied arbitrary heat transfer coefficient (h_c_) values ranging from 0 to 12 W·m^−2^·K^−1^, as depicted in Figure 6. In this calculation, the ambient temperature was set to 300 K, consistent with the field test temperature measurements. As shown in Figure 6, the cooling capacity of the AHYOSPs reached 103.77 W/m^2^, whereas the highest capacity of NYO was 77.41 W/m^2^, consistent with the field test results, thus suggesting that the proposed strategy for assembling nanoparticles was successful.

## 4. Conclusions

This study demonstrated the significant potential of assembled hollow yttrium oxide spherical particles (AHYOSPs) to enhance passive radiative cooling (PRC) compared to nano-sized yttrium oxide (NYO) and baseline polydimethylsiloxane (PDMS). Using micro-sized hollow yttrium oxide particles, this research showed an increase in both solar reflectance and emissivity within the atmospheric window. Specifically, the AHYOSPs exhibited a solar reflectance of 73.72% and an emissivity of 91.75%, representing a substantial improvement over the cooling efficiency of NYO and the PDMS baseline.

Field tests further validated our theoretical models, demonstrating that the AHYOSPs achieved the lowest temperatures during daylight due to their superior solar reflectance. At night, the differences between the AHYOSPs and NYO narrowed, indicating that emissivity played a more crucial role in their cooling performance under reduced solar radiation.

Ultimately, this study confirms that the structural control of yttrium oxide particles is pivotal for optimizing the cooling capabilities of PRC films. These findings suggest that further enhancements in particle assembly and morphology could lead to more efficient passive cooling solutions, potentially reducing our reliance on energy-intensive cooling methods and contributing to efforts to combat global warming.

## Figures and Tables

**Figure 1 polymers-16-01636-f001:**
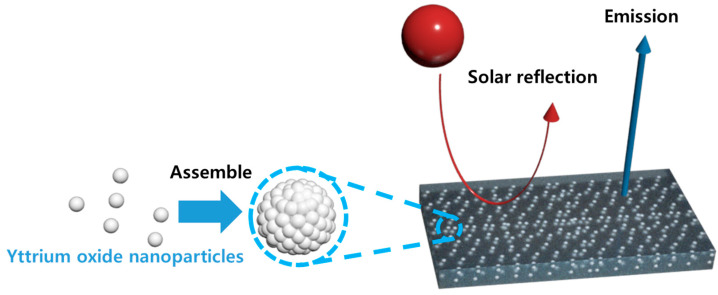
Schematic image of PRC film mechanism.

**Figure 2 polymers-16-01636-f002:**
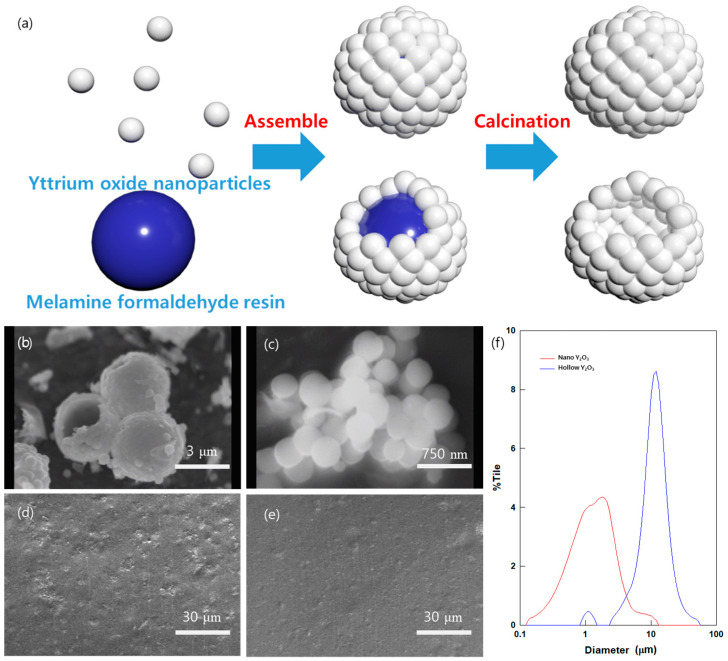
(**a**) Schematic of AHYOSP synthesis. SEM image of PRC particles: (**b**) AHYOSPs and (**c**) NYO. Cross-sectional images of (**d**) AHYOSPs and (**e**) NYO films. (**f**) DLS analysis graph of PRC particles.

**Figure 3 polymers-16-01636-f003:**
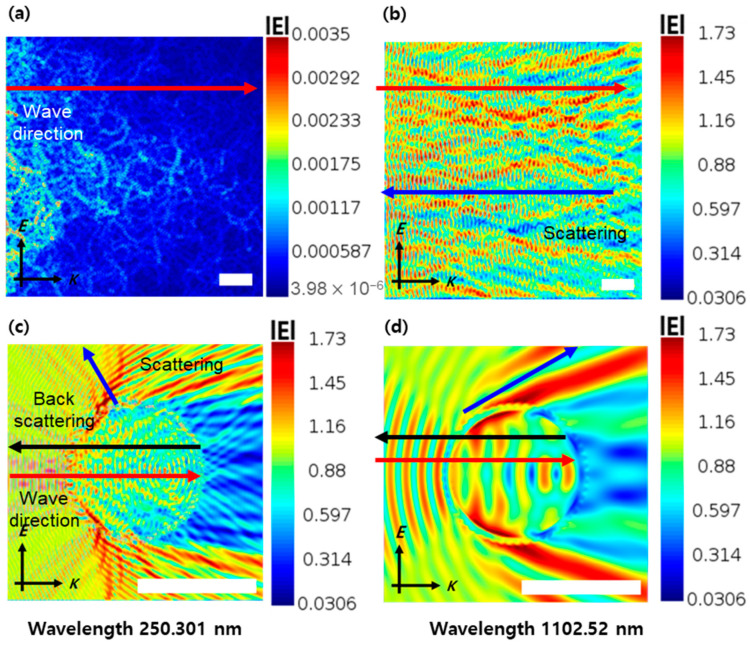
Electric field distribution of Y_2_O_3_ particles. (**a**,**b**) NYO and (**c**,**d**) AHYOSPs; wavelength scale bar: 3 μm.

**Figure 4 polymers-16-01636-f004:**
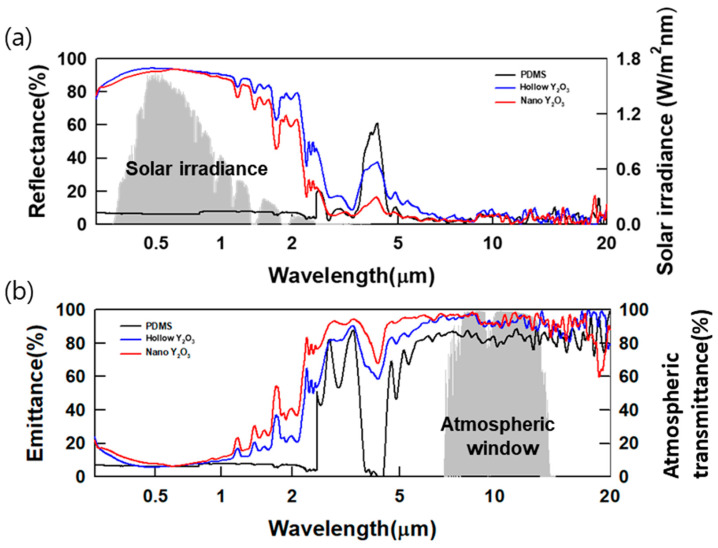
Optical properties of PRC films: (**a**) reflectance and (**b**) emittance.

**Figure 5 polymers-16-01636-f005:**
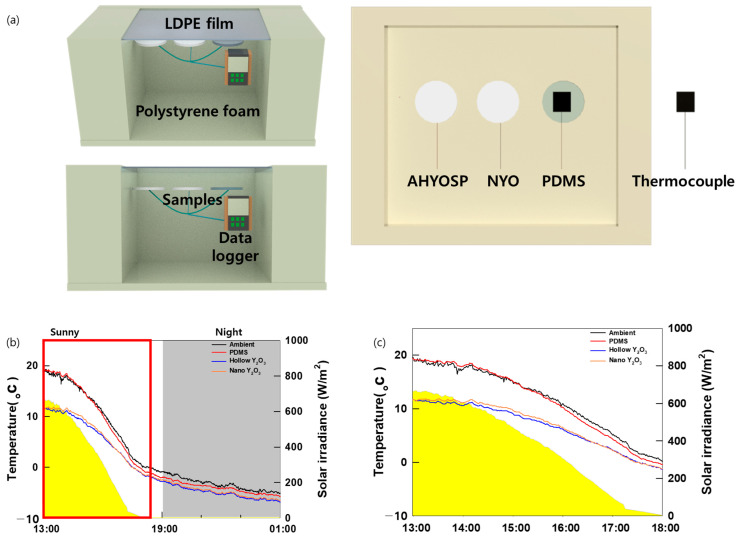
(**a**) Schematic illustration of the field test setup. (**b**) Field test temperature graph. (**c**) Field test temperature graph during the daytime.

**Figure 6 polymers-16-01636-f006:**
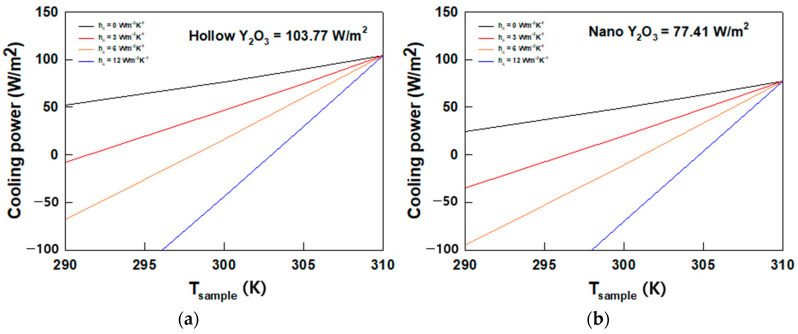
Cooling power at ambient temperature 310 K: (**a**) AHYOSPs and (**b**) NYO.

## Data Availability

Data available upon request (due to privacy).

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
