# Peer review of "Assembly of Hollow Yttrium Oxide Spheres from Nano-Sized Yttrium Oxide for Advanced Passive Radiative Cooling Materials"

_polymers, 2024, doi:10.3390/polym16121636_

Round 1

Reviewer 1 Report

Comments and Suggestions for Authors

The research article is devoted to a relevant topic -new material for passive radiative cooling. The presented results are relevant, original, and practically significant. However, the quality of data presentation can be improved. Please consider the following comments.

1. All equipment and software used should be listed in the Experiment section. The manufacturer of the equipment and the conditions for conducting the analysis should also be indicated there.

2. In the Experiments it is mentioned that the sizes of the melamine-formaldehyde cores can be adjusted by changing the experimental parameters. More detailed information is not provided. It is advisable to present it in the text, with a description of what concentrations are optimal. At least the concentrations used by the authors should be given.

3. In the work, the dimensions of both nano-sized yttrium oxide and assembled hollow yttrium oxide spherical particles were determined by dynamic light scattering method. The method for hollow yttrium oxide spherical particles synthesizing is presented in the article, but it is not clear how yttrium oxide particles were obtained (by synthesis or by destruction of spheres) before determining the size by this method. It is not very clear what DLS was used for. More accurate and clear information was obtained using the SEM method.

3. The microphotographs in the figures 2b and 2c are informative and allow one to distinguish the spheres and their structure. But it is difficult to understand how these spheres are located on the surface in the figures 2d and 2e. A higher magnification may be required.

Minor point-

- The numbering of the figures is incorrect; it is not related to the sequence of appearance in the text. E.g. Figure 2.a is referred at chapter 2.2, line 95, but placed at chapter 3.1. line 211.  

- Related to the above: Field test chapter is split into 2.5 and 3.4, thus Figure 5(a) is referred right after Figure 1.

Comments on the Quality of English Language

The English language is appropriate and understandable. Minor editing required.

Misprints

Line 249 Figure 4.b trancmittance - transmittance

Line 003 Please double check that the last word "Materi- als" is not split per two lines

Line 055 BaSO4 - subsript for '4'

Line 211 Figure 2.f legend: consider using same case, i.e. both "Nano ... Hollow ..." cf. Figure 4 and others.

Reviewer 2 Report

Comments and Suggestions for Authors

The manuscript (polymers-3047925) represents the preparation of passive radiative cooling (PRC) using assembled hollow yttrium oxide spherical particles (AHYOSPs), and also investigates the cooling efficiency. It is a good work. The reviewer suggests that the manuscript can be considered for publication in this journal after minor revision. The following question should be responded in the revised manuscript.

1) In Figures 1 and 2, the white small ball should be labeled as yttrium oxide nanoparticles, while the blue ball should be labeled as melamine formaldehyde resin.

2) 3.1 Morphology of Y2O3 Particles should be changed into 3.1 Assembling of hollow yttrium oxide spherical particles.

3) Some wrong sentences and formatting errors in this paper need to improve.

Comments on the Quality of English Language

No comments.
